# GLYCOGYM: BENCHMARKING GLYCAN PROPERTY PREDICTION

## ABSTRACT

Glycan property prediction is an increasingly popular area of machine learning research. Supervised learning approaches have shown promise in glycan modeling; however, the current literature is fragmented regarding datasets and standardized evaluation techniques, hampering progress in understanding these complex, branched carbohydrates that play crucial roles in biological processes. To facilitate progress, we introduce GlycoGym, a comprehensive benchmark suite containing six biologically relevant supervised learning tasks spanning different domains of glycobiology: glycosylation linkage identification, tissue expression prediction, taxonomy classification, tandem mass spectrometry fragmentation prediction, lectin-glycan interaction modeling, and structural property estimation. We curate tasks into specific training, validation, and test splits using multi-class stratification to ensure that each task tests biologically relevant generalization that transfers to real-life glycan property prediction scenarios. We benchmark a diverse range of approaches to glycan representation learning, spanning fingerprint-based baselines, language models operating on IUPAC-condensed sequences, and graph neural networks explicitly designed for glycan topology, including Sweet-Net, GLAMOUR, and the recent GIFFLAR architecture. We find that specialized glycan encoders consistently outperform simple baselines for the more complex tasks. GlycoGym will help the machine learning community to focus their efforts on scientifically relevant glycan prediction problems and will be regularly updated through integration with the glycowork Python package. Toward this end, all data and code used to run these experiments will be made available at GitHub and Zenodo.

## 1 INTRODUCTION

Glycans are composed of monosaccharides, such as glucose or galactose, joined via glycosidic linkages in up to 20 different configurations. This then results in branched, nonlinear structures, which extend from one reducing end (typically covalently linked to either a protein or lipid backbone) to multiple non-reducing ends that are exposed to the outside of the cell (Varki, 2016). Examples of glycan function include the extravasation of leukocytes, by binding to the Lewis X epitope (Brazil et al., 2016). Furthermore, in mammalian milk, glycans are known to activate immune cells and act anti-pathogenic via competitive inhibition of viral/microbial proteins (Jin et al., 2023; 2025).

Predicting interactions and other properties of complex carbohydrates or glycans is a major challenge in current glycoinformatics (Bojar & Lisacek, 2022). Hampered by both an inherent sequence diversity and complexity, as well as sparsity and heterogeneity of available data, glycan-focused machine learning still lags behind models for other biological sequences, such as proteins. Recent work on extracting information from glycan sequences has shown that important biological properties, such as the interaction of glycans with lectins (their protein receptors), can be predicted (Lundstrøm et al., 2022; Joeres & Bojar, 2024), which is crucial to uncover host-pathogen interactions and find new cancer-targeting proteins (Bojar et al., 2021; Nieto-Fabregat et al., 2024; Xu et al., 2024b). This presents a potential opportunity to understand the functions of glycans at scale, which is a major unsolved problem.

Since the application of machine learning methods to glycan sequences presents a relatively recent development (Bojar et al., 2020; 2021; Bojar & Lisacek, 2022), it is still unclear to what extent

glycan encoders learn general vs task-specific features. We thus argue that, analogous to standard practices for other biomolecules (Rao et al., 2019; Ren et al., 2024), new glycan models should be evaluated on a broad array of different property prediction tasks, to obtain a representative view of their performance and applicability to common tasks in the glycosciences. We argue that this will not only settle the question of learning general vs task-specific features but also aid in model development and advance current glycoinformatics approaches.

However, such benchmarking presents a challenge in glycan-related tasks, as data generation in glycobiology is arduous and expensive, coupled with the lack of centralized storage for curated data. To fill this gap and to facilitate computational biologists in assessing their models on standardized datasets, including defined splits, we here present a comprehensive suite of curated datasets for various glycan property prediction tasks. In addition to updated datasets for existing tasks, we also present two new such tasks with associated curated datasets, predicting glycan tissue expression and reconstructing glycan fragmentation during mass spectrometry. Overall, we show that current glycan encoders extract meaningful information from glycan sequences, and outperform competitive baselines on more complex tasks, yet emphasize that bespoke glycan-AI architectures will be needed in the future to reach even higher performance and spur glycoinformatics applications.

## 2 DATA

Over time, many datasets have been collected that contain glycan properties. Typically, these are created once, but not maintained afterwards. In 2024, Xu et al. published a benchmark using a collection of these datasets, but it has not been updated since then, despite the included datasets having grown since (Xu et al., 2024a).

GlycoGym updates the GlycanML datasets and also extends to new datasets and domains of glycan property prediction. Multiple groups have independently introduced some of the datasets presented in this work. Over several years, the BojarLab manually curated and extended these data from the glycobiology literature, integrating the datasets into the glycowork Python package (Thomès et al., 2021). To solve the maintenance problem of previous efforts, we will regularly update the datasets and distribute them to the community.

### 2.1 DATASETS

In the following, we will describe each dataset and its significance in glycobiology. Table 1 provides information about each dataset after preprocessing as described in Section 2.2.

#### 2.1.1 GLYCOSYLATION LINKAGE IDENTIFICATION

Glycans can be categorized by their biosynthetic history, whether they are N-linked or O-linked to a glycoprotein, part of a glycolipid, or present in a free form, such as in breast milk (Jin et al., 2023). While this task is easily decidable for many glycans and a trained human expert (such as categorizing chitobiose-terminated glycans as N-linked), it is a good baseline to check whether a model has a basic understanding of glycan sequence distributions. To simulate a human expert labeling the data, we utilize the `get_class` method from `glycowork` as a rule-based proxy and an easy baseline for predicting the glycosylation of glycans.

#### 2.1.2 TISSUE EXPRESSION PREDICTION

Individual tissues express typical glycans, such as brain-specific gangliosides (e.g., GT1b and GQ1b) (Huang et al., 2022). The tissue expression dataset collects such findings for tissues and cell lines, providing a multilabel classification task over these data. One caveat of this dataset is that the question of whether a certain glycan exists in a tissue or not is unfalsifiable, as it would require a negative existence proof. The fact that a glycan has not been found in a particular tissue does not necessarily mean that it is truly absent. Therefore, a negative label in this dataset can have one of two meanings: (i) a tissue does not express a certain glycan, or (ii) the expression has not been observed yet.

Initially, this dataset has been labeled with 271 different UBERON (Mungall et al., 2012), Cell Ontology (CO) (Diehl et al., 2016), and NCBI identifiers (Schoch et al., 2020). To simplify this

Table 1: Key properties of the presented datasets. The average glycan size was calculated across the entire dataset.

| Dataset | Task | avg. glycan size #monosacchs/glycan | split sizes train / val / test |
|---------|------|-------------------------------------|--------------------------------|
| Linkage | 5-class | 7.97 | 8,402 / 2,024 / 999 |
| Tissue | 20-label | 6.75 | 1,603 / 361 / 181 |
| Kingdom | 13-label | 6.09 | 11,932 / 2,826 / 1,419 |
| Spectrum | regression | 5.37 | 108,195 / 29,191 / 14,867 |
| *Lectin-glycan interaction modeling* | | | |
| random | regression | 4.94 | 104,417 / 30,379 / 15,204 |
| cold-lectin | regression | 4.96 | 107,085 / 28,436 / 14,479 |
| cold-glycan | regression | 4.94 | 107,040 / 28,749 / 14,211 |
| Structure | node-feat. reg. | 5.90 | 4,295 / 1,153 / 572 |

dataset, we aggregated labels along the UBERON ontology tree into 35 classes representing the most meaningful groups. After the general preprocessing described below, the number of classes was further reduced to 20.

### 2.1.3 TAXONOMY CLASSIFICATION

In the same way that glycans can be tissue-specifically expressed, they can be taxon-specifically expressed, such as the expression of core a1-3 fucosylated N-glycans in plants and invertebrates (Strasser, 2016). Here, expression differences can occur at all taxonomic levels. Therefore, we provide versions of this for all eight levels, namely Domain, Kingdom, Phylum, Class, Order, Family, Genus, and Species. For the benchmark, we recommend using the Kingdom dataset, as it is the most informative, both glycobiologically and from a classification point of view. Therefore, we will only use this in the following and refer to it as *the* taxonomy dataset.

Similarly to the tissue expression dataset, all taxonomy datasets have the caveat that a negative label can either mean that (i) a taxon does not produce a certain glycan, or (ii) it has not been observed yet, which is especially problematic for shallowly investigated taxa.

In many previous works, the taxonomy datasets have been presented as single-label classification tasks. Yet a single glycan can be found in multiple species if it is conserved. Therefore, a perfect, deterministic model cannot achieve perfect performance metrics because it is incapable of resolving the ambiguity of individual glycans. To overcome this problem, we follow the practice established by GIFFLAR and pose the taxonomy datasets as multi-label classification tasks (Joeres & Bojar, 2024).

### 2.1.4 TANDEM MASS SPECTROMETRY FRAGMENTATION

This dataset is taken from the publication of CandyCrunch (Urban et al., 2024). Urban et al. provided 489,103 MS/MS spectra of glycans, annotated with structures. One of the main preprocessing steps in the machine learning scheme was to bin the list of masses and intensity peaks into a real-valued vector and normalize it. To achieve this, they defined 2,048 intervals of 1.4454 Da between 39.741 Da and 3,000 Da, and the intensities in each bin were summed. These vectors were then converted into unit vectors. Starting from their preprocessing steps, we provide two datasets: (i) the development dataset contains 152,253 MS/MS-spectra of 652 different glycans, while (ii) the deployment dataset contains all 489,103 spectra from 3,391 glycans. This split into two datasets enables researchers to quickly develop new model architectures with moderate memory and computational demands on the development dataset and to publish final models on the deployment dataset.

Because measuring MS/MS spectra is highly dependent on experimental setups (e.g., the ion mode, glycan derivatization, or fragmentation energy), the geological location, and even the air pressure, the same glycan molecule can have vastly different spectra (Han & Costello, 2013). This poses an exceptionally complex task for deep learning models.

### 2.1.5 LECTIN-GLYCAN INTERACTION MODELING

The lectin-glycan interaction prediction dataset arguably presents the most interesting task for practitioners. Interactions between lectins (carbohydrate-binding proteins) and glycans are complex biochemical processes characterized by highly specific binding patterns. Lectins bind only to certain glycan motifs, and minor changes can turn a glycan into a non-binder. Therefore, the performance of a prediction model not only depends on the quality of the glycan encoder, as in all other tasks presented here, but also on a protein encoder.

Most protein-ligand interaction prediction models and LectinOracle (Lundstrøm et al., 2022), the state-of-the-art model in this specific field, utilize the classic Y-shaped protein-ligand interaction prediction model architecture. This combines separate lectin and glycan encoders whose embeddings are concatenated and fed into a prediction head. In Joeres & Bojar (2024), the authors demonstrated that embeddings from the ESM2-t33 model yield the best results, among the four compared state-of-the-art protein encoders. Therefore, for this dataset, we pre-embedded all protein sequences using ESM2-t33 and used them as a fixed input, alongside a trainable glycan encoder.

The goal of this dataset is thus not to produce the best overall model to predict lectin-glycan interactions, but to estimate how well a particular glycan encoder module in this Y-shaped architecture extracts relevant information for binding to lectins, compared to others.

### 2.1.6 STRUCTURAL PROPERTY ESTIMATION

Recently, Thomès et al. published an analysis of structural data for glycans, including a section on machine learning models to predict structural glycan features, such as torsion angles and solvent-accessible surface area (SASA) (Thomès et al., 2025). This dataset contains monosaccharide-level information and therefore requires a model architecture that produces embeddings for monosaccharides and the glycosidic bonds between them. In the same paper, the authors proposed a variant of the SweetNet model (Burkholz et al., 2021) to provide exactly these predictions. They reached accuracies that are within the experimental inaccuracies, even in out-of-distribution settings.

Similar to the fragmentation prediction, one IUPAC string can have multiple correct solutions here, as a single glycan can have multiple conformations. Therefore, an optimal model needs to generate probability distributions for the angles between monosaccharides. For this work, we reproduced the dataset creation script and introduced a validation set for hyperparameter tuning and comparison.

## 2.2 PREPROCESSING

Each dataset underwent several preprocessing steps as visualized in Figure 1. Firstly, we canonicalized all IUPAC-condensed strings using the Universal Input Parser (Urban et al., 2025) within the `glycowork` package to avoid redundancies in branch ordering and human errors in sequence annotation. To make the data accessible to models operating on atomic fingerprints or graphs, we then translated the canonicalized IUPACs into SMILES using GlyLES (Joeres et al., 2023). As a side effect, this step filtered out all glycans containing floating elements (i.e., uncertain topology) or wildcards (i.e., uncertain sequence), as those uncertainties cannot be represented in SMILES.

For the three classification datasets (Taxonomy, Tissue, and Linkage), we removed all classes with fewer than 15 entries. Predicting these sparse properties otherwise becomes unstable and only offers limited insight into the model's understanding of that property. Similarly, we removed all glycans with fewer than 15 spectra from the fragmentation dataset.

For the two largest datasets, fragmentation prediction and interaction prediction, we provide two datasets. First, a development dataset with ∼150,000 data points to allow for fast testing and hyperparameter optimization of new models, and second, the complete datasets for deployment. For the fragmentation prediction dataset, downsampling was performed by randomly sampling spectra from glycans with more than 1000 spectra in the dataset to reduce redundancy and computational load. The downsampling of the interaction dataset was done purely randomly over the list of interactions.

Lastly, we provide complex data splits for the presented datasets. Each dataset consists of 70% training data, 20% validation data, and 10% test data for accurate performance estimation. For the classification data, we ensured that each class was represented in each of the data splits using multiclass stratification. For fragmentation prediction and the GlyContact dataset (Thomès et al., 2025),

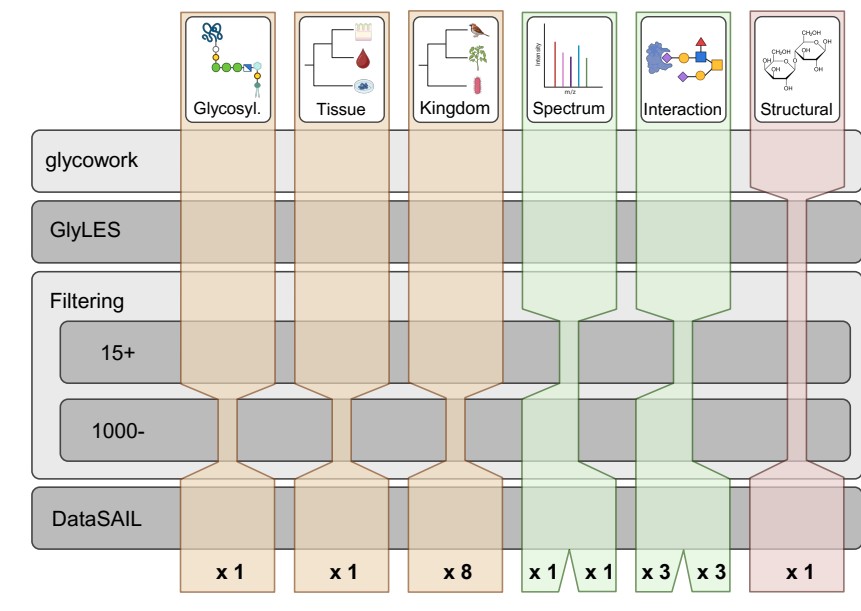

Figure 1: Visualization of the preprocessing steps for each dataset. Where the dataset column narrows, the corresponding step did not affect the dataset. If only one side is narrowing, this indicates that the resulting development dataset was affected, but not the deployment dataset. The splitting of a column at the bottom indicates that we provide a development and a deployment version of this dataset. For some fields, such as taxonomy or interaction predictions, we provide multiple datasets to serve different tasks. This is indicated by the numbers at the bottom. Classification datasets are colored orange, regression datasets are colored green, and structural predictions are colored red. Figure created with BioRender.

we ensured that all spectra or 3D structures, respectively, of one glycan were in the same split. For the interaction data, we provide a random split, a cold-lectin split, and a cold-glycan split to enable testing of a model in scientifically relevant scenarios. In the cold splits, the main focus is to test how well a model can generalize to unseen lectins or glycans, i.e., out-of-distribution performance, while the random split tests for in-distribution performance. All described data splits were computed using DataSAIL (Joeres et al., 2025).

## 3 MODELS

To evaluate the current state-of-the-art on our new benchmark, we evaluated 11 different model architectures and three statistical baselines. This provides a broad survey of models and shows performance improvements over simple baselines. Details about the training setup, architectural specifics, used packages, and hardware are given in Section A.1.

### 3.1 FINGERPRINT-BASED MODELS

As baseline machine learning models, we tested Random Forests (Breiman, 2001), Support Vector Machines (Vapnik, 2013), Gradient Boosting (Friedman, 2001), and Multilayer Perceptrons (Werbos, 1974) operating on 1,024-bit Morgan Fingerprints based on SMILES strings of the glycans (see Figure 2b). These fingerprints were computed using RDKit(Landrum et al., 2006).

### 3.2 IUPAC LANGUAGE-BASED MODELS

The oldest group of models we will compare in the field of glycan property prediction comprises language-based glycan encoders, operating on the IUPAC-condensed representation (see Figure 2c). The first glycan language model in this group was SweetTalk (Bojar et al., 2020), which utilized

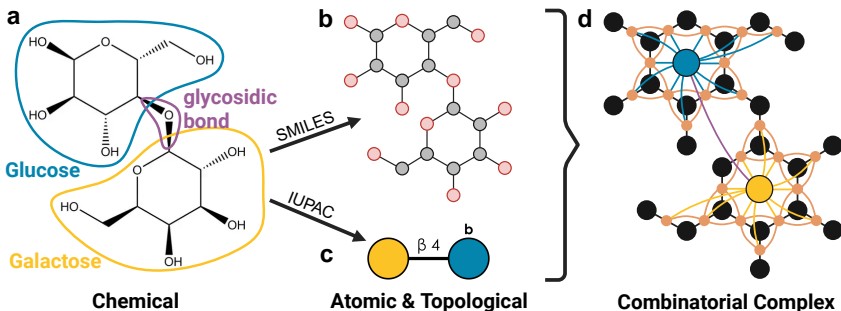

Figure 2: Visualization of different ways to represent the glycan Lactose. The models in this work use various representations for the same molecule.

LSTMs to extract a feature representation from the IUPAC-condensed sequence. For this work, we reimplemented the model based on the architectural decision made by Bojar et al.

With the rise of LLMs and their increasing accessibility, researchers have also applied them to IUPAC-condensed sequences. Two models have been published based on the BERT architecture (Devlin et al., 2019), namely glyBERT (Dai et al., 2021) and SweetBERT (Rubia-Rodríguez et al.). However, we were unable to run glyBERT on our datasets, and for SweetBERT, we were unable to locate the code for the model. Therefore, we mention them for completeness, but unfortunately cannot compare their performance on this new benchmark.

### 3.3 GEOMETRIC DEEP LEARNING-BASED MODELS

In the third group of models, we evaluate geometric deep learning (GDL) models. These are typically graph neural networks (GNNs) of various complexity, operating on a multitude of different graph representations of glycans (see Figure 2b-d).

#### 3.3.1 SWEETNET

The earliest model in this category was SweetNet by Burkholz et al. (2021), which was the first GNN trained on graph representations of glycans and was primarily designed for glycans. It is the first model to utilize the specific polymeric structure of glycans (see Figure 2c), as it operates on a monomer graph with two types of nodes: nodes for monosaccharides and nodes for glycosidic bonds between them. Initial features for all nodes are random vectors from an embedding space indexed by the entity a node represents. This architecture proved helpful in various tasks, such as estimating structural properties.

#### 3.3.2 GLAMOUR

A similar architecture is used in GLAMOUR (Graph Learning over Macromolecule Representations) (Mohapatra et al., 2022), which was published for general polymers. Similar to SweetNet, GLAMOUR models have nodes representing monomeric units. Yet these are not connected through another type of node, but via featurized edges. The features of the nodes and edges are Morgan Fingerprints of the monomers, and of the bond-forming atoms of the edges between them, respectively. Applied to glycans, GLAMOUR graphs store monosaccharides in the nodes and glycosidic bonds in their edges. As evaluated in the GIFFLAR publication, we used the MPNN backend within the GLAMOUR framework as it performed best in their analysis (Joeres & Bojar, 2024).

Out of the box, GLAMOUR only provides end-to-end trained classification and regression models, but not the encoder alone. Therefore, we only applied it to end-to-end tasks such as glycosylation or fragmentation prediction, but not lectin-glycan interaction prediction or structural property prediction.

### 3.3.3 GNNGLY

The publication of GlyLES created the opportunity to translate IUPAC sequences into SMILES and made it possible to represent glycans as atomic graphs (see Figure 2b). This was picked up in GNNGLY (Alkuhlani et al., 2023), where simple graph convolutional layers were applied to atomic graphs to predict glycan properties. Since we could not find a codebase for GNNGLY either, we reimplemented it as well as possible, following the descriptions from the paper.

### 3.3.4 RELATIONAL GCN

In the predecessor work, GlycanML, the Relational GCN (RGCN) (Schlichtkrull et al., 2018) performed best across most of their benchmarks. Because of its ability to deal with heterogeneous graphs, we applied it to the same glycan graph representation as the GIFFLAR model presented below (see Figure 2d). Both use the higher-order message passing scheme developed by Hajij et al. (2022). Their only difference is the type of graph convolution applied. The RGCN utilizes relational graph convolutional layers, whereas GIFFLAR employs graph isomorphism layers Xu et al. (2018).

### 3.3.5 GIFFLAR

The last model we compare is GIFFLAR (Joeres & Bojar, 2024), the most recently published model for deep learning on glycans, heralding new state-of-the-art performances on some of the datasets presented here.

GIFFLAR introduced a new representation of glycans: the combinatorial complex. While this graph construction was known to mathematicians and had been recently applied to molecular property prediction (Bodnar et al., 2021a;b), GIFFLAR is the first to transfer it to glycan representation learning. It represents the glycan as a combinatorial complex with three ranks of nodes: (i) atomic nodes, (ii) nodes representing bonds between atoms, and (iii) nodes representing monosaccharides (marked in Figure 2d in black, orange, and blue/yellow, respectively). For these graph complexes, GIFFLAR defined three types of within-rank edges, e.g., atom-to-atom, and two upward-rank edges, namely atom-to-bond, and bond-to-monosaccharide. Extensive details on the mathematical background are given in the original publication. As described for the RGCN, GIFFLAR applies heterogeneous higher-order message-passing on these graphs using graph isomorphism layers.

## 4 RESULTS

In Table 2, we present the performance of the models on the classification tasks and spectrum prediction. The glycosylation linkages are predicted almost perfectly by all models, also clearly outperforming the rule-based baseline `get_class`. The reason why GNNGLY and RGCN report an MCC of 0 is that the MCC becomes undefined when the model never predicts a certain class. The tissue dataset poses a much more challenging task. Models with a focus on the topological structure of glycans perform slightly better than others (see SweetNet, SweetTalk, and GIFFLAR, which have topological features as part of the input). But overall, fingerprint-based methods perform similarly to advanced deep learning models. The same phenomenon can be observed for the Kingdom prediction: advanced deep learning models perform only marginally better than simple baselines. However, reaching an MCC of 0.8 indicates that the models understand the data and correlation better than for tissue prediction.

For the fragmentation prediction, we did not test most of the fingerprint-based models, as it is computationally too demanding. Additionally, some architectures would require fitting multiple models, thus not understanding the relations between different intensity peaks. The models we evaluated learned some notion of the spectra, but similarly to the structural property modeling task, one glycan has multiple different spectra; therefore, the improvements gained by the deterministic architectures we tested were minimal, as they cannot predict the variety of possible outcomes. Furthermore, as outlined above, the chemical structure investigated is only one of many factors that influence the results of a tandem mass spectrometry analysis.

In the lectin-glycan interaction prediction task, we investigated three standard, glycobiologically interesting settings: (i) a random split, (ii) a cold-lectin split, and (iii) a cold-glycan split. The two cold splits ensure that no lectin or glycan, respectively, had interactions in more than one split; therefore,

Table 2: Performances measured by MCC, and Cosine-Similarity on four benchmark datasets. The number of parameters is measured for the glycosylation linkage prediction task.

| Model | #trainable params [M] | Glycosylation | Tissue | Kingdom (Tax.) | Spectrum |
| --- | --- | --- | --- | --- | --- |
| | | *Matthews Corr. Coef. (MCC)* ↑ | | | *Cosine-Sim.* ↑ |
| get_class | n.a. | 0.784 | n.a. | n.a. | n.a. |
| *ECFP4-based models* | | | | | |
| RF | – | 0.932 | 0.499 | 0.649 | – |
| SVM | – | 0.940 | 0.473 | 0.770 | – |
| XGB | – | 0.950 | 0.523 | 0.806 | – |
| MLP | 1.1 | 0.943 | 0.516 | **0.822** | 0.3454 |
| *IUPAC-based language models* | | | | | |
| SweetTalk | 3.3 | 0.906 | 0.518 | 0.715 | -0.293 |
| *Monomer-based GDL models* | | | | | |
| SweetNet | 2.3 | 0.947 | **0.544** | 0.779 | 0.4504 |
| GLAMOUR | 2.5 | 0.963 | 0.442 | 0.812 | 0.2035 |
| *All-atom GDL models* | | | | | |
| GNNGLY | 0.5 | 0 | 0.447 | 0.654 | **0.5083** |
| RGCN | 2.7 | 0 | 0.481 | 0.711 | 0.0201 |
| GIFFLAR | 2.3 | **0.983** | 0.501 | 0.817 | 0.4728 |

Table 3: MSE comparison of glycan encoders for lectin-glycan interaction prediction (lower is better). The prediction of the mean baseline is the mean label of the interacting molecules in the training set. Therefore, it serves as a maximum-memorization baseline. $\text{mean}_{\text{global}}$ always predicts the mean label of the entire train dataset for any input.

| Model | #trainable params [M] | random | cold-lectin | cold-glycan |
| --- | --- | --- | --- | --- |
| *Statistic baseline* | | | | |
| mean | n.a. | 0.8685 | 0.8206 | 1.0532 |
| $\text{mean}_{\text{global}}$ | n.a. | 0.8814 | 0.8171 | 1.0194 |
| *Fingerprint-based encoders* | | | | |
| ECFP4 | 4.6 | 0.7105 | 0.6217 | **0.9936** |
| *IUPAC-based encoders* | | | | |
| SweetTalk | 5.5 | 0.7602 | 0.6668 | 1.0351 |
| *Monomer-based GDL encoders* | | | | |
| SweetNet | 4.8 | 0.7503 | 0.6413 | 1.0496 |
| *All-atom GDL encoders* | | | | |
| GNNGLY | 2.6 | 0.7762 | 0.6663 | 1.0453 |
| RGCN | 4.7 | 0.7008 | 0.6384 | 1.0434 |
| GIFFLAR | 4.3 | **0.6862** | **0.6175** | 1.0402 |

they allow for analyzing a model's ability to generalize to unseen lectins or glycans. Naturally, cold-splits pose much more challenging prediction tasks, as ML models tend to memorize what they have seen during training rather than generalize based on the properties of the data. Notably, in Table 3 all models perform much better on the cold-lectin split than on any other split. This indicates that ESM embeddings generalize better to new lectins than the learned glycan embeddings generalize to new glycans, as observed in the cold-glycan split comparison. Additionally, simple mean-based baselines are outperformed by more complex models, except for the cold-glycan split, which poses an especially challenging task.

On the structural property estimation dataset, the Von Mises-SweetNet model clearly outperforms the baselines. This can be mostly attributed to the architecture of this model, as it does not directly predict values but rather the parameters of a probability distribution from which values can be sam-

Table 4: Performances on structural data as RMSE (lower is better). Shown are predictions for disaccharide torsion angles ($\phi, \psi, \omega$), solvent-accessible surface area (SASA), and flexibility (flex).

| Model | $\phi$ [°] | $\psi$ [°] | $\omega$ [°] | SASA [$\mathring{A}^2$] | flex [$\mathring{A}$] |
|---|---|---|---|---|---|
| *Statistic baselines* | | | | | |
| Mean | 24.4428 | 48.6437 | 36.5432 | 35.2529 | 0.8060 |
| Median | 23.7085 | 34.7275 | 22.91 | 35.3106 | 0.7140 |
| *GDL encoders* | | | | | |
| SweetNet | 20.2274 | 29.3752 | 24.8013 | 19.2676 | 0.5014 |
| von Mises-SweetNet | **6.3792** | **11.0938** | **8.7365** | **14.8314** | **0.3963** |

pled, which aligns with the physiological reality of multiple conformational states that a glycan can inhabit.

## 5 DISCUSSION

Overall, the goal of our work here is to provide an assessment of the current state-of-the-art in glycan-focused deep learning, with the most comprehensive set of glycan tasks and datasets. Furthermore, the aim of GlycoGym is also to evaluate future model architectures, gain an understanding of their relationship to existing models, and identify tasks for which they may be particularly well-suited. Our group will update the datasets used here, ensuring an increase in data quality and quantity for the tasks here over time.

While our current approach for fragmentation spectrum prediction can be useful for focusing on a given experimental setting, we argue that a general solution to this task will need non-deterministic models, similar to the structural property dataset (Thomès et al., 2025), to accommodate multiple valid answers given the same glycan sequence input. For this, predicting the parameters of a Gaussian mixture model (instead of directly predicting intensities) could be a promising direction.

Especially for the taxonomy task, we suspect that the composition of a glycan is already very informative for a coarse-grained classification into kingdoms, as the alphabet used for glycan sequences is highly non-uniform and taxonomically gated Srivastava et al. (2020). This could explain why simpler models that do not consider topology still perform well on this task here.

We note that current glycan encoders appear to yield models for lectin-glycan interaction prediction that generalize better to new lectins than to new glycans. This indicates room for improvement on the side of the glycan encoder to leverage the rich information in complex carbohydrates more effectively. It has been shown previously that lectins require a specific 3D context of their binding motifs (Thomès et al., 2025), which is potentially why purely sequence-based encoders do not yet capture this information. Furthermore, lectins often have long-range/distal requirements and/or restrictions for binding Bojar et al. (2022), which may also not be efficiently captured in current architectures.

We point out that no model performed best in all tasks, highlighting the need to select the appropriate method/model for a specific scientific question, which is facilitated by our benchmark. Overall, we conclude that there is a need for both a more diverse set of glycan property prediction tasks as well as new model architectures that simultaneously (i) consider the branched nature of glycans, with effects that are long-range in sequence but not necessarily in 3D space, and (ii) accommodate probabilistic outputs, to reflect the highly dynamic nature of glycans in solution. Different glycan conformations may have different biological effects (e.g., in the interaction with lectins), and new state-of-the-art glycan encoders will need to incorporate this characteristic to achieve the long-anticipated potential of this information-rich biological sequence.

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

## A  APPENDIX

### A.1  TRAINING SETUP

All preprocessing is implemented in `python` v3.11. For the preprocessing, we used `glycowork` v1.6.2 and `glyles` v1.2.2. The models and training are implemented as `pytorch-lightning` (v2.5.1) modules using `torch` 2.7.1 and `torch-geometric` v2.6.1. All geometric deep learning (GDL) models have eight hidden layers with 256 dimensions. The SweetNet has 16 layers, roughly matching the number of parameters of the other models, and has the same depth of view as GIFFLAR, which can examine monomers eight steps away. For all GDL models, we use a single-layered feed-forward network with 128 hidden neurons as the prediction head. The only exception from these specifications is GLAMOUR; for this model, we adapted the original git repository to our needs. We did not tune the hyperparameters for any of these models; instead, we present them with commonly used parameterizations.

All training sessions were conducted on a machine equipped with 128 CPUs, 1 TB of CPU RAM, and an NVIDIA V100 with 16GB of GPU RAM.

