# OpenReview forum: "GlycoGym: Benchmarking Glycan Property Prediction"
_ICLR.cc/2026/Conference — ICLR 2026 Conference Withdrawn Submission_

### Official Review · Reviewer_EoDK · 2025-10-28

**Soundness:** 2
**Presentation:** 2
**Contribution:** 2
**Rating:** 4
**Confidence:** 4

**Summary:**

Despite their biological significance, glycans remain underexplored within the AI/ML community. This study tries to bridge the gap by develop an extensive benchmark suite for machine learning research. It is a great effort that should be encouraged. However, glycans are involved in a wide range of biological processes, and hence their properties are complex. The coverage of this benchmark is limited. The contributions of the presented benchmark suite should be elaborated more.

**Strengths:**

Try to develop a comprehensive benchmark suite spanning different domains of glycobiology.

**Weaknesses:**

* Glycosylation identification is a simple task and does not require machine learning.
* Two tasks (tissue expression prediction and taxonomy classification) are "unfalsifiable"
* Other datasets are already available and ML-ready. What are the contributions of this work beyond collecting those datasets and trying existing models.
* Biological/chemical justification is needed for the proposed training, validation, and test splits in the tandem mass spec fragmentation prediction task. Same is required for other tasks.
* GlycoNMR (https://data.mlr.press/assets/pdf/v01-11.pdf) is closely relevant.

**Questions:**

None

---

### Official Review · Reviewer_V66d · 2025-10-30

**Soundness:** 3
**Presentation:** 3
**Contribution:** 2
**Rating:** 4
**Confidence:** 4

**Summary:**

This paper introduces GlycoGym, a comprehensive benchmarking suite for glycan property prediction, encompassing six biologically relevant tasks: glycosylation linkage identification, tissue expression prediction, taxonomy classification, tandem mass spectrometry fragmentation prediction, lectin-glycan interaction modeling, and structural property estimation. The authors curate and preprocess datasets, provide standardized splits, and benchmark a wide range of models including fingerprint-based methods, IUPAC-based language models, and graph neural networks (e.g., SweetNet, GLAMOUR, GIFFLAR). They find that specialized glycan encoders outperform simpler baselines on complex tasks, though no single model dominates across all tasks.

**Strengths:**

1. This paper provides a comprehensive and biologically relevant benchmark, which covers six diverse and biologically meaningful tasks, providing a holistic evaluation of glycan representation learning.
2. This paper uses canonicalization, SMILES conversion, and multi-class stratification. Provides development/deployment splits and cold splits for OOD evaluation.
3. This work has extensive Model Comparison. GlycoGym evaluates 11 architectures and 3 baselines, spanning fingerprints, language models, and GNNs, offering a broad perspective on current capabilities.

**Weaknesses:**

1. This benchmark does not include fully atomistic models (e.g., Equivariant GNNs, Force Field-based models), which could better capture 3D conformational dynamics.
2. The study lacks evaluation of pre-trained glycan models (e.g., masked language modeling, graph pre-training) or transfer learning models, which have shown success in proteins and small molecules.
3. Incomplete coverage of recent architectures. Models like Transformers for graphs (e.g., GraphGPS, GraphFormers) or geometric GNNs (e.g., SE(3)-GNNs) are not included, despite their potential for glycan modeling.
4. The benchmark does not assess whether models can leverage multiple tasks or modalities (e.g., sequence + structure) for improved generalization.

**Questions:**

1. Have you considered incorporating pre-trained glycan models (or other biological models) into this benchmark?
2. How do you plan to extend GlycoGym to include 3D structural data or dynamics simulations for glycans?
3. Could you discuss the potential of multi-task learning or cross-model architectures for glycan property prediction?
4. What are your plans for updating the benchmark with new tasks or datasets (e.g., glycan-protein interaction networks) ?

---

### Official Review · Reviewer_EyhD · 2025-11-02

**Soundness:** 3
**Presentation:** 4
**Contribution:** 2
**Rating:** 2
**Confidence:** 4

**Summary:**

This paper introduces and dataset and then benchmarks existing models in representation learning / property prediction tasks in glycans, a class of biomolecules

GlycoGym introduces a new benchmark suite for glycan property prediction, aggregating six supervised tasks across glycobiology (e.g. glycosylation linkage identification, tissue-specific glycan expression, taxonomy classification, tandem MS fragmentation, lectin–glycan binding, and glycan structural feature prediction). The authors curate these datasets with careful splits (stratified training/validation/test).

A  range of machine learning models – from simple molecular fingerprints and sequence-based language models to specialized glycan graph neural networks (SweetNet, GLAMOUR, GIFFLAR) – are benchmarked on each task. Based on the results, the paper concludes that domain-specific glycan encoders significantly outperform baseline representations on the more complex prediction problems, underscoring the importance of tailored representation learning for these branched biomolecules. All data and code are openly released (integrated with the glycowork package) to facilitate community adoption and continuous updates to the benchmark.

**Strengths:**

Strong benchmark: This work fills a clear gap by providing a unified,  benchmark for glycan property prediction (and a promise to maintain it?). The tasks cover diverse aspects of glycoscience.
tasks: GlycoGym consolidates existing a number of prediction tasks (e.g. glycan taxonomy classification, lectin–glycan binding) that were already in use and commonly accepted. It also introduces less-explored problems like tissue-specific glycan expression and mass spectrometry fragmentation prediction (which was also already published). It broadens prior benchmarks .

The benchmark compares a wide array of representation-learning approaches on equal footing – from simplistic fingerprint vectors to sequence-based models and advanced glycan-specific GNNs (such as SweetNet, GLAMOUR, and the recent GIFFLAR architecture). This exhaustive baseline evaluation is technically valuable, revealing how traditional featurizations and modern deep learning models fare on the same tasks. It provides clear evidence that specialized glycan encoders extract substantially more relevant information for complex tasks than generic baselines, a useful insight for practitioners designing models for glycans.

**Weaknesses:**

Incremental: The contribution is largely in assembling and standardizing datasets, without introducing new modeling techniques or fundamentally new prediction tasks - no additional experimental work is reported or collated. Notably, a very similar benchmark GlycanML (Xu et al., 2024) was recently released with 11 glycan learning tasks (including taxonomy, immunogenicity, glycosylation type and protein–glycan interaction prediction). The submission does not make a strong case to differentiate itself from this prior effort. Authors should clarify unique aspects (e.g. inclusion of structural and MS/MS tasks) and discuss differences from GlycanML to establish novelty. Many of the tasks and data in GlycoGym are drawn from existing glycoinformatics studies, meaning the paper’s primary contribution is in curation and evaluation rather than new discovery. For example, the lectin–glycan interaction dataset leverages a standard protein–ligand model setup from prior work (e.g. using the LectinOracle architecture and pre-trained protein embeddings), and the glycan structural feature prediction task is directly based on a recent published dataset and model (torsion angle/SASA predictions from Thomes et al., 2025).

Only supervised prediction: The benchmark focuses exclusively on single-task supervised learning, without exploring broader machine learning avenues that could be relevant to glycans. For instance, no generative modeling or inverse design task is included, despite the interest in designing or synthesizing glycans with desired properties. Likewise, the authors do not investigate multi-task learning or transfer learning across their tasks. This is a missed opportunity: given that many glycan tasks are related, a joint training approach might improve performance (as shown in prior work, where multi-task training on glycan taxonomy boosted other glycan property predictions). Incorporating or at least discussing such approaches would have strengthened the paper’s methodological contribution and relevance to advanced molecular ML techniques.

As a pure benchmark paper, the analysis sometimes stops at reporting metrics, without delving into why certain models succeed or fail. The results indicate that simple baselines approach the performance of complex GNNs on some classification tasks, which could imply those tasks are not very challenging or that current models are not exploiting glycan structure effectively – but the discussion on these points is limited.

While glycans are biologically important, the work’s direct impact might be confined to the small (but growing) glycoinformatics community. The benchmark’s breadth is excellent for glycan specialists, but its relevance to broader molecular ML could be emphasized more.

I think Nature Scientific Data is a better venue for work like this rather than an AI conference.

**Questions:**

why is only one reference given to the breast milk origins of some glycans but not others ?

"Glycans can be categorized by their biosynthetic history, whether they are N-linked or O-linked to a glycoprotein, part of a glycolipid, or present in a free form, such as in breast milk (Jin et al., 2023)."

---

### Official Review · Reviewer_9MMQ · 2025-11-06

**Soundness:** 2
**Presentation:** 3
**Contribution:** 3
**Rating:** 4
**Confidence:** 3

**Summary:**

This paper introduces GlycoGym, a benchmark created to help researchers evaluate how well machine learning models can predict properties of glycans, which are complex sugars involved in many biological processes. Authors included six tasks that reflect real scientific challenges, namely figuring out how glycans are linked, where they are found in the body, which organisms produce them, how they behave in mass spectrometry, how they interact with proteins, and what their structures look like. The authors carefully curated each dataset, using biologically meaningful train, validation, and test splits. They tested a wide variety of models, including traditional machine learning approaches, language models that read glycan sequences, and graph-based deep learning models like GIFFLAR, SweetNet, and GLAMOUR.

**Strengths:**

a.	This task is a biologically rich but still relatively unexplored area in machine learning. GlycoGym addresses this by providing the first carefully built and modern benchmark dedicated to predicting glycan properties. In doing so, it brings clarity and focus to a growing yet underdeveloped corner of bioML research.
b.	The benchmark is comprehensive. It covers several types of tasks: classification, regression, and structured prediction. They are all tied directly to real-world biological problems. By including both in-distribution and out-of-distribution settings, GlycoGym tests how models perform not just on familiar data, but also when pushed into new challenges.
c.	It’s clear that the authors put a lot of care into constructing the dataset and evaluation process. They use thoughtful split strategies, like multi-class stratified and biologically inspired cold splits. This is to make sure the setup mirrors practical research scenarios. Moreover, preprocessing is done using reliable tools like GlyLES and DataSAIL. The benchmark also evaluates a wide range of models, from classic machine learning algorithms like Random Forest and XGBoost, to modern language models and advanced graph-based architectures. This diversity makes GlycoGym genuinely useful for both ML researchers and domain scientists looking for a strong, ready-to-use resource.

**Weaknesses:**

a.	The benchmark excludes important models like glyBERT and SweetBERT because of missing code or compatibility issues. Although this makes sense, it does restrict how much of the current landscape the benchmark covers. To allow for a more thorough comparison in subsequent iterations, the authors should either re-implement these models or get in touch with the original authors.
b.	The authors note that they chose to use widely used defaults rather than perform hyperparameter tuning.  Although this decision encourages equity, models that are known to be sensitive to hyperparameter settings, such as MLPs or RGCNs, may unintentionally suffer as a result.  Where possible, it would be better to use the hyperparameters from the original publications rather than depending on generic defaults.  A simple sensitivity analysis could be used to evaluate each model's robustness if those are not available.  It is challenging to assert a fair performance comparison across approaches without such considerations, which is a fundamental objective of any benchmark.

**Questions:**

None

---

### Note · Authors · 2025-11-25

**Comment:**

We want to thank the reviewers for their time and extensive feedback. Unfortunately, we are unable to prepare a sufficient revision of our manuscript in the given time, but we will use the input for a future publication of this work.

**Withdrawal Confirmation:**

I have read and agree with the venue's withdrawal policy on behalf of myself and my co-authors.